# Empowering Rehabilitation: Design and Structural Analysis of a Low-Cost 3D-Printed Smart Orthosis

**DOI:** 10.3390/polym16101303

**Published:** 2024-05-07

**Authors:** Florin Popișter, Mihai Dragomir, Paul Ciudin, Horea Ștefan Goia

**Affiliations:** Department of Design Engineering and Robotics, Faculty of Industrial Engineering, Robotics and Production Management, Technical University of Cluj-Napoca, B-dul Muncii 103-105, 400641 Cluj-Napoca, Romania; ciudin.oc.paul@student.utcluj.ro (P.C.); goia.io.horea@student.utcluj.ro (H.Ș.G.)

**Keywords:** 3D-printed parts, structural analysis, complex shapes, 3D printing technologies, structural integrity

## Abstract

Three-dimensional (3D) printing of polymer materials encompasses a wide range of applications and innovations. Polymer-based 3D printing, also known as additive manufacturing, has gained significant attention due to its versatility, cost-effectiveness, and potential to revolutionize various industries. The current paper focuses on obtaining a durable low-cost rehabilitation knee orthosis. Researchers propose that the entire structure should be obtained using modern equipment within the additive manufacturing domain—3D printing. The researchers focus on determining, through a 3D analysis of the entire 3D model assembly, which parts present a high degree of stress when a kinematic simulation is developed. The entire 3D model of the orthosis starts based on the result obtained from a 3D scanning of the knee joint of a patient, providing a precise fixation, and allowing for direct personalization. Based on the results and identification of the critical parts, there will be used different materials and a combination of 3D printing strategies to validate the physical model of the entire orthosis. For the manufacturing process, the researchers use two types of low-cost fused filament fabrication (FFF), which are easy to find on the worldwide market. The motivation for manufacturing the entire assembly using 3D printing techniques is the short time in which complex shapes can be obtained, which is relevant for the present study. The main purpose of the present research is to advance orthotic technology by developing an innovative knee brace made of 3D-printed polymers that are designed to be lightweight, easy-to-use, and provide comfort and functionality to patients during the rehabilitation process.

## 1. Introduction

Present research is focused on contributing to the fields of wearable technology, assistive devices, and rehabilitation. The development of a knee orthosis system that has real-time corrective feedback, offering solutions for enhancing gait and posture, is the result that authors are focused on. The practical implementation of the results is valuable for individuals with gait-related difficulties and provides healthcare and rehabilitation professionals with useful tools.

Rehabilitation is often a lengthy and challenging process. Patients may find it difficult to fully regain control and strength in their limbs. Many individuals in today’s world suffer from injuries that limit their mobility and range of motion. Therefore, assistive devices can play a crucial role in restoring these patients’ independence.

Knee injuries and conditions, such as ligament tears, osteoarthritis, and postoperative recovery, often require targeted rehabilitation exercises to restore proper joint function and improve patient outcomes. It is important to ensure that patients perform these exercises correctly and consistently, as improper movements can hinder progress and even cause further injury.

The knee joint is crucial for daily activities and is the largest and most injured joint in the human body. It comprises four major components, namely the bones, cartilage, ligaments, and tendons. Injuries to the ligaments or arthritis can cause significant pain for many individuals [1].

Numerous studies have been conducted on developing smart, easy-to-use knee orthoses [2,3,4,5,6,7,8,9]. However, few have focused on the patients’ motivation to correct their movement. By utilizing electrical muscle stimulation (EMS), patients can accurately identify when an incorrect movement has been performed, thus encouraging them to correct the error. The 3D-printed knee orthosis is equipped with embedded IMU (Inertial Measurement Unit) sensors that measure and analyze joint orientation, acceleration, and rotation. This precise movement data can identify deviations from prescribed movement patterns or improper joint alignments. Additionally, strategically located EMG (Electromyography) sensors on the surrounding leg muscles provide real-time feedback on muscle activation levels, enabling the evaluation of muscle coordination during knee movements.

The production of orthoses using conventional methods, such as metal bending or plastic molding injections, is a time-consuming process that requires more than one manufacturing equipment. Additionally, the shapes and dimensions of the orthosis must be meticulously adjusted to fit the patient’s body. Moreover, creating multiple personalized orthoses of the same quality is challenging, and incorporating intricate designs is often necessary. Recently developed three-dimensional (3D) printing technology allows for the precise design of orthoses using computer-aided design (CAD) programs, mitigating the disadvantages of conventional methods. This technology enables the creation of structures that are difficult to implement manually [2,3,4]. There are also polymers on the market that present a high level of resistance from a mechanical proprieties point of view, but the production cost is high [10,11,12].

The performance of 3D-printed polymer components has evolved, enabling their use in structural and rigorous applications. As the price of 3D printing has decreased, additively generated orthotics have become a viable alternative to conventionally constructed orthotics. Early prototypes were large and heavy due to the technology of the time, which consisted of enormous motors and driving systems. 3D-printed braces and orthoses are transforming the field of orthotics by offering personalized solutions that are more comfortable and functional. The ability to customize designs to fit individual anatomy ensures a precise fit and improved patient compliance. Owing to the adaptability of 3D printing technology, these lightweight devices can include complex geometries and features. Material selection allows for the appropriate mechanical properties, while rapid prototyping enables quick iterations and modifications [13,14,15,16,17,18,19,20].

The aim of this research is to contribute to the advancement of orthotic technology by developing an innovative knee brace made of 3D-printed polymers that are easy to find on the market, PLA (polylactic acid) and rPET (polyethylene terephthalate). The orthosis is designed to be lightweight and easy-to-use, providing tailored support, comfort, and functionality to patients during the rehabilitation process. The objective of this investigation was to examine the behaviors of two constituent elements within the entirety of the ensemble, identified by the authors as experiencing elevated levels of stress and strain. In pursuit of this objective, the study concentrated on the analysis and assessment, through comparative means, of the optimal strategies and efficacy provided by the two commercially accessible materials.

## 2. Prototype Design and Manufacture

The prototypical iteration has been designed to address the exigencies associated with knee joint rehabilitation after injury, a critical phase wherein the regulated mobilization of the patient’s lower extremity plays a pivotal role in fostering optimal recuperative outcomes.

The smart orthosis architecture encompasses a bespoke orthotic structure, predicated upon the three-dimensional mesh representation of the patient’s knee joint, alongside a comprehensive array of constituent components including a control mechanism, sensory apparatus, and feedback actuation system. The tailored orthosis is fabricated utilizing additive manufacturing techniques, notably 3D printing, to ensure optimal conformity and comfort for individuals convalescing from injury. The sensory instrumentation comprises two Inertial Measurement Unit (IMU) sensors, specifically the MPU6050 model, strategically situated at the upper and lower extremities of the orthotic assembly to accurately ascertain the angular parameters of the knee joint. Furthermore, an Electromyography (EMG) sensor, exemplified by the H124SG model, is employed to monitor the musculature engagement pertinent to knee joint activation. The control system is realized through the integration of a System on a Chip (SoC) development board, typified by the Beetle ESP32, selected for its compact form factor, robust computational capabilities, and proficient Wi-Fi and Bluetooth connectivity attributes. The feedback actuation mechanism is constituted by a Transcutaneous Electrical Nerve Stimulation (TENS) apparatus, incorporating pads seamlessly integrated within the orthotic structure to ensure optimal contact with the superior and inferior knee articulations of the patient. The feedback provision is effectuated through the application of mild electrical stimuli for brief temporal intervals in instances where joint kinematics exceed predefined thresholds.

The knee orthosis mechanical 3D design process involved the utilization of a 3D model derived from the patient’s knee joint anatomy. Employing a Creaform Go Scan 50, a digital rendition of the knee and leg was meticulously crafted, presented in Figure 1, serving as the foundation for the orthotic design, tailored to exact specifications. Utilizing structured light technology, the 3D scanner meticulously captured a sequence of images, yielding a highly precise and detailed portrayal of the limb’s surface morphology. Employing a multifaceted approach, the limb underwent scanning from diverse vantage points and orientations to ensure comprehensive coverage and fidelity of the captured data. Subsequently, the acquired images underwent meticulous processing utilizing VXelements 2020 software, facilitating the generation of a three-dimensional representation of the patient’s limb. Notably, the Ver.1 software afforded capabilities for artifact removal and flaw rectification, thereby guaranteeing the attainment of a superior-quality rendition of the limb, integral to the subsequent orthotic design process.

The limb underwent scanning and thus generated a mesh (see Figure 2) representation which subsequently translated into a solid body, forming the foundation for the knee orthosis fabrication process. This transformation from mesh to solid was facilitated through the utilization of CATIA V5, a prominent 3D modeling software prevalent in engineering and manufacturing domains. Within CATIA V5, the mesh data underwent conversion into a solid entity, wherein surface thickness parameters were delineated and uniformly applied across the entirety of the limb. This resulted in the generation of a solid component meticulously aligned with the contours delineated during the scanning procedure.

The inception of a prototype, as presented in Figure 3, and the formulation of an optimal design, created in CATIA V5 software, represented a pivotal juncture within this project. Armed with a thorough comprehension of the project’s objectives and associated challenges, conceptualizations and visions gradually concreted into tangible manifestations. The iterative process and experimental endeavors assumed a critical role in discerning and honing the most suitable design trajectory for the endeavor. Acknowledging that the pursuit of innovation frequently entails traversing through multiple iterations and embracing a predisposition towards exploring diverse avenues facilitated the realization of this research endeavor.

Harnessing the inherent capabilities of 3D printing technology, the prototype facilitated a hands-on evaluation and appraisal of the system’s operational efficacy. This critical phase afforded the opportunity for the positioning and integration of components to be scrutinized, thereby ensuring seamless functionality and adherence to ergonomic standards. A meticulous analysis ensued, enabling precise adjustments and enhancements to be implemented in pursuit of the desired outcome. The 3D-printed prototype (see Figure 4) assumed a pivotal role in the evaluation of the joint system’s performance, particularly in terms of articulatory fluidity and precise alignment of the knee joint. This evaluative phase served as a discerning mechanism for identifying potential areas necessitating refinement or rectification in the initial design, thereby facilitating the incorporation of requisite modifications in subsequent iterations. Beyond functional assessment, the prototype underwent rigorous mechanical testing to gauge its durability, resilience, and overall structural robustness. This entailed subjecting the orthotic brace to controlled stresses, tensions, or impacts to ascertain its capacity to withstand the rigors of everyday usage without compromising performance or posing a risk of injury.

### 2.1. Orthosis 3D Model

SolidWorks 2022 was chosen as the CAD design phase software to create the final proposed device (see Figure 5). Solidworks provided a comprehensive set of tools and features that enabled precise and meticulous design detailing. Due to its advanced features and capabilities, the design process involved multiple iterations and extensive testing to ensure the best possible device was created. From the positioning of components to the overall ergonomics and user experience, every aspect was subjected to meticulous analysis. The challenge of striking the right balance between form and function required careful consideration. The goal was to create a device that was not only visually appealing, but also offered optimal performance and comfort. Bearings are still included in the final design of the project, similar to the approach taken in the first iteration. The use of bearings offers significant benefits in terms of facilitating smooth motion and reducing friction within the device. The bearings selected are 6808 2RS. The ability of the 6808 2RS bearings to effectively handle both radial and axial loads was one of the key factors in their selection. This was due to the dynamic nature of the knee joint and the various forces and movements to which it is subjected. The 6808 2RS bearings were found to have the required load capacity, allowing the orthosis to withstand the complex forces while maintaining reliable and seamless motion.

The aim of this research was to study the behavior of two of the components (see Figure 6) of the whole assembly of the orthosis which, in the authors’ opinion, were subject to the highest levels of stress and strain. The two components define the hinge of the orthosis. In this respect, the research focused on analyzing and determining, by comparison, the most effective strategy and the effectiveness offered by two materials readily available on the market.

### 2.2. 3D Printing Processes

To ensure optimal functionality and structural integrity, the design of the orthosis required careful consideration of the mechanical components and their calculations. Two distinct polymer matrices, specifically PLA (polylactic acid) and rPET (polyethylene terephthalate), were evaluated, encompassing two commercial brands for each polymer variant. The study was concentrated upon two types of materials since are easy to buy, cheap, and based on the properties (see Table 1) of the considered polymers [21], are suitable for the entire orthosis.

The 3D printing process was executed employing two specialized apparatuses that were readily accessible and commercially available, Creality Ender 3 Pro/Creality Ender 3 S1 Pro, as shown in Figure 7.

In the context of 3D printing, tolerances denote the permissible degree of variance or fluctuation observed in the dimensions of a printed object in relation to its specified design parameters. A tolerance specification of 0.03 mm (or 30 microns) signifies that the dimensions of the printed object have the potential to deviate by up to 0.03 mm from the intended measurements. Such a tolerance level is regarded as exhibiting a relatively high degree of precision within the realm of 3D printing and is deemed suitable for a diverse array of applications. It is pertinent to acknowledge that achieving close tolerances can be influenced by a multitude of factors, encompassing printer calibration, filament quality, intricacy of design, and the specific printing parameters employed.

During the 3D printing phase of the orthosis, a diverse array of methods and tools were employed to optimize fabrication processes and attain the desired outcome. Among these methodologies (see Figure 8), 3D printing infill patterns, Grid, Triangles, and Gyroid, pertained to the utilization of tree supports (turquoise color in Figure 8), a functionality accessible slicing software platform like Cura (Version 5.6.0). Additionally, supplementary techniques were employed to augment the printing procedure and bolster the overall quality of the resultant orthotic structure. For the printing outcomes within the present study, using rPET and PLA material (see Figure 9, Figure 10 and Figure 11), it was used for the printing-process-encompassing parameters (see Table 2) such as layer height, print speed, and temperature settings, to ensure optimal print quality and adherence to design specifications.

### 2.3. Analisys of the Lateral Thigh Pivotal Components

Following the finalization of the design and manufacturing, an extensive analysis was conducted to assess the mechanical performance of the lateral thigh support component. This particular element was chosen for analysis due to its pivotal role in furnishing stability and support to the user’s thigh.

The mechanical test methodolgy employed was focused on the hinge element of the orthosis and provided through the use of a specialized bending measuring equipement, Instron 3366—Dual Column Tabletop Models (Instron Inc., Norwood, MA, USA). The entire scope of this was to identify which infill patern form the 3D process handles forces in a better manner. Further aspects of the methodology are presented below.

The selection of materials, namely rPET (polyethylene terephthalate) and PLA (polylactic acid), was influenced by their distinct mechanical properties, as presented in Table 1, which exerted significant influence over the analysis. Notably, the analysis that has been developed is characterized by a comparatively high Young’s modulus, indicative of its inherent stiffness and resistance to deformation.

Studies regarding mechanical testing of components or assemblies of parts obtained through 3D printing technologies were conducted and considered to be relevant in the context of polymer parts with complex shapes [22,23,24].

The lateral femoral support underwent rigorous examination through a multitude of load scenarios and boundary conditions within the investigation, enabled the simulation of real-world operational conditions, affording invaluable insights into the behavior of the support component when subjected to diverse forces and stresses. Through this process, an assessment of the structural integrity, strength, and deformation characteristics of the component were analyzed, thereby informing critical decisions pertaining to its optimization and performance enhancement.

Figure 12 illustrates a photograph capturing the sample during the bending test using the Instron 3366, Dual Column Tabletop Models (Instron Inc., Norwood, MA, USA). Notably, the grips employed in the testing apparatus for both the orthosis and its constituent fragments were fabricated using 3D printing technology. The primary criterion guiding their selection was to guarantee the steadfast stability of the designated sample throughout the duration of the measurements.

Figure 13, Figure 14 and Figure 15 showcase photographic representations of the sample undergoing the bending test. The grips utilized in the testing apparatus for the orthosis pivotal assembly parts, crafted through 3D printing methodologies using rPET and PLA, are instrumental in upholding the stability of the designated sample throughout the testing procedures. The efficacy of these components is evaluated based on their capacity to sustain stability during the testing process.

The force–displacement data indicate variations among the three methods utilized in this study regarding the two components comprising the hinge of the prototype knee orthosis. The infill pattern employed in 3D printing technology for fabricating these components demonstrates disparate mechanical stiffness, particularly between rPET and PLA polymers materials. According to the mechanical characteristics provided by the 3D printing material manufacturer, PLA exhibits a higher degree of plasticity. Notably, when employing rPET material, it is observed from Figure 16 and Figure 17, that the curves are nearly as steep, up to approximately 100 N. Specifically, when utilizing the rPET material with a Grid hinge infill, it withstood forces up to 170 N, while the second Gyroid pattern infill exhibited resistance up to 198 N. Conversely, in the case of the Triangles hinge infill, it sustained forces up to 148 N.

In the second test, the joint consisted of components obtained through 3D printing utilizing PLA as the material. The joint was subjected to testing with the same three types of infill patterns, yielding more satisfactory results. When components were fabricated with a Grid infill, the Flexure load at Tensile Strength recorded a value of 300 N. Utilizing a Gyroid infill pattern resulted in recorded forces reaching a value of 379 N, whereas the employment of a Triangles infill pattern yielded forces measuring at 232 N. All values are presented in Table 3 and Table 4.

All the values of the strength of the “hinge” assembly obtained from the testing process of the whole batch of specimens are directly influenced by the printing process. A particularity of this unconventional process is the adhesion of the coating using polymer materials. Layer adhesion refers to the cohesive bonds formed between consecutive layers of material during the additive manufacturing process, ensuring the structural integrity and stability of the printed object [25,26,27,28].

## 3. Discussions

This study is dedicated to examining the mechanical characteristics of the components comprising the hinge assembly of knee orthosis. Utilizing two cost-effective polymer materials, namely rPET and PLA, readily accessible in the global market, the study employs the unconventional method of 3D printing to fabricate these components. The primary objective of this investigation is to ascertain the maximum values of flexure load and tensile strength that the two components can withstand. Notably, within the scope of this study, three infill pattern methods selected that are well known for their durability or that are common and used in code generation applications for 3D printing and implemented using freely available applications for the physical production of polymer parts. The authors assert the relevance of this information within the contextual framework of the study.

Following the analysis and identification of materials, as well as the strategic evaluation of infill pattern effectiveness and efficiency within the context of the study’s theme, the focus persists on the physical realization and testing of a Low-Cost 3D Printed Smart Orthosis (see Figure 18).

One of the foremost challenges encountered in the project pertained to the creation of a system capable of furnishing real-time corrective feedback to the user predicated upon IMU sensor data. This necessitated the development and deployment of algorithms adept at precisely discerning and analyzing joint angles, delineating thresholds indicative of correct movement, and triggering the relay system to effect corrective measures. The integration of real-time feedback epitomized a pioneering approach towards ameliorating gait dynamics and postural alignment through wearable technology.

A significant hurdle encompassed the integration and calibration of multiple sensors, including the IMU MPU 6050 and EMG sensor. Sensor fusion methodologies were employed to amalgamate data streams from these sensors, thereby furnishing precise and dependable measurements requisite for detecting aberrant movement patterns and muscle activation. The calibration of sensors to ensure the consistency and accuracy of readings posed an additional layer of complexity, necessitating iterative adjustments and validation endeavors to attain optimal operational efficacy.

The development of a tailored mobile application utilizing MIT App Inventor, coupled with the establishment of robust Bluetooth communication between the application and the knee brace system, posed a substantial challenge. The application served as a user-centric interface, facilitating the real-time display of sensor data, management of system settings, and seamless user interaction. Effectuating Bluetooth communication protocols, safeguarding data integrity, and crafting an intuitive application interface entailed a fusion of software development acumen and user experience design prowess. This innovative feature markedly enhanced the usability and accessibility of the knee orthosis system (see Figure 18) rendering it more appealing to prospective users.

## 4. Conclusions

The project’s conclusions underscore the successful attainment of the proposed objectives while underscoring the consequential impact of the undertaken endeavors. The outcomes stemming from the development and deployment of the knee orthosis system featuring real-time corrective feedback signal its potential to enhance gait dynamics and postural alignment, thus making substantive contributions to the domain of wearable technology and assistive devices.

The utilization of 3D printing technology ensures a bespoke fit tailored to the wearer, augmenting comfort levels while concurrently streamlining production timelines and reducing costs vis-à-vis conventional orthotic manufacturing methodologies. The advent of this intelligent knee brace holds paramount significance as it addresses the exigency for more efficacious and streamlined rehabilitation modalities, catering to individuals recuperating from knee-related injuries or surgical interventions. Conventional therapeutic interventions, such as physiotherapy and pharmacotherapy, often entail monotony and demoralization for patients, coupled with prohibitive socio-economic barriers rendering them inaccessible to certain demographics.

The amalgamation of the knee orthosis system into clinical practice harbors profound implications across diverse disciplines encompassing rehabilitation, physiotherapy, and athletic training. The system’s real-time feedback mechanism coupled with its corrective functionalities evinces potential avenues for augmenting mobility, averting injury occurrences, and refining overall movement kinematics.

This initiative has successfully conceptualized and implemented a knee brace system featuring real-time corrective feedback, thereby enriching the realm of pervasive technology and assistive devices. The discerned outcomes elucidate the system’s promise in ameliorating gait dynamics and postural alignment. While acknowledging existing limitations, continued research and developmental endeavors hold promise for surmounting these impediments and laying the groundwork for future applications in the domains of rehabilitation, physiotherapy, and athletic training. This endeavor is not only consequential but also holds transformative potential in enhancing the quality of life and mobility for individuals grappling with gait-related challenges.

As for the future work, the iteration of the prototype is intended to be implemented in a specialized rehabilitation clinic, from a kinesiology point of view. Also, the use of this prototype will be conducted under the supervision of a doctor and a therapist and from the patients’ point of view, it will be tested with people of different ages.

## Figures and Tables

**Figure 1 polymers-16-01303-f001:**
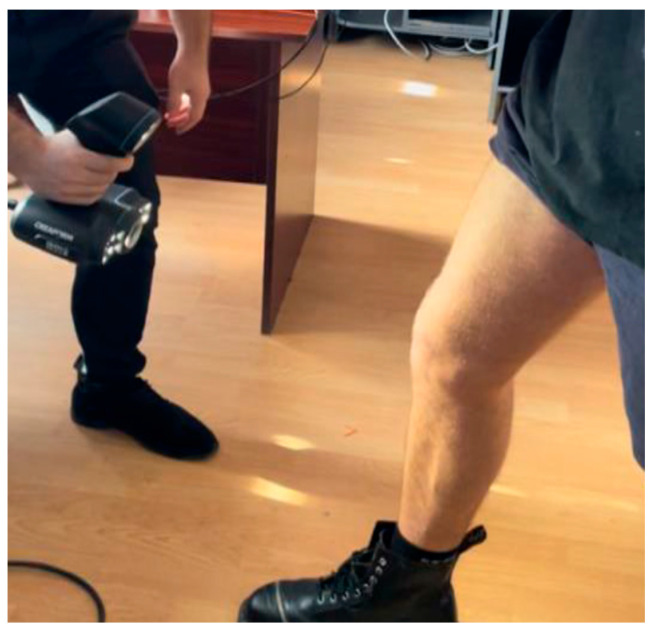
Scanning of the leg using a 3D Scanner.

**Figure 2 polymers-16-01303-f002:**
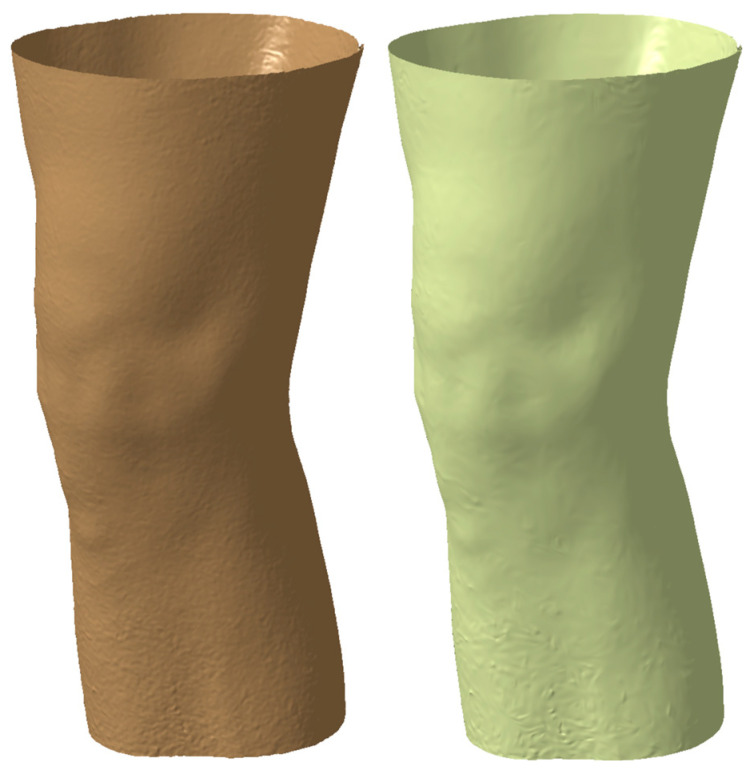
3D mesh after processing the 3D scanning process and the generated 3D surface.

**Figure 3 polymers-16-01303-f003:**
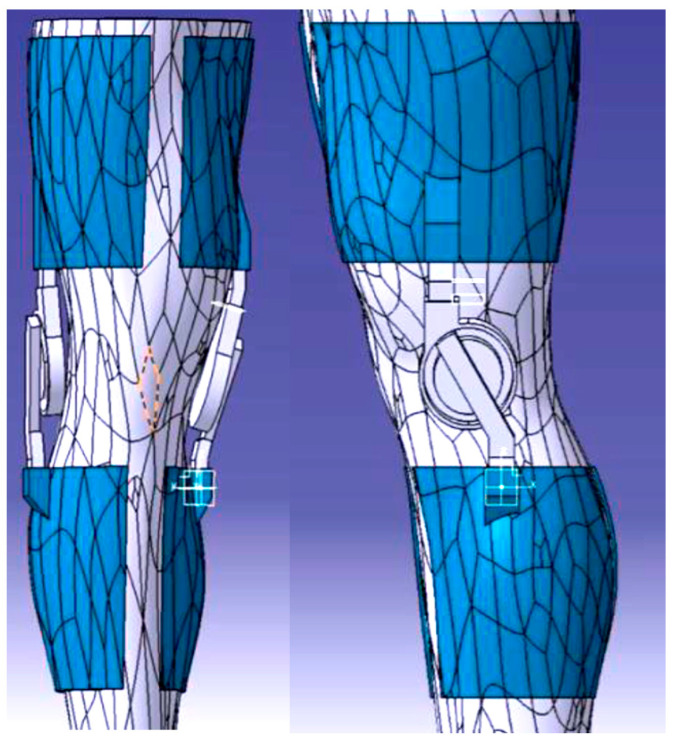
3D model of the first concept of the hinge of the orthosis.

**Figure 4 polymers-16-01303-f004:**
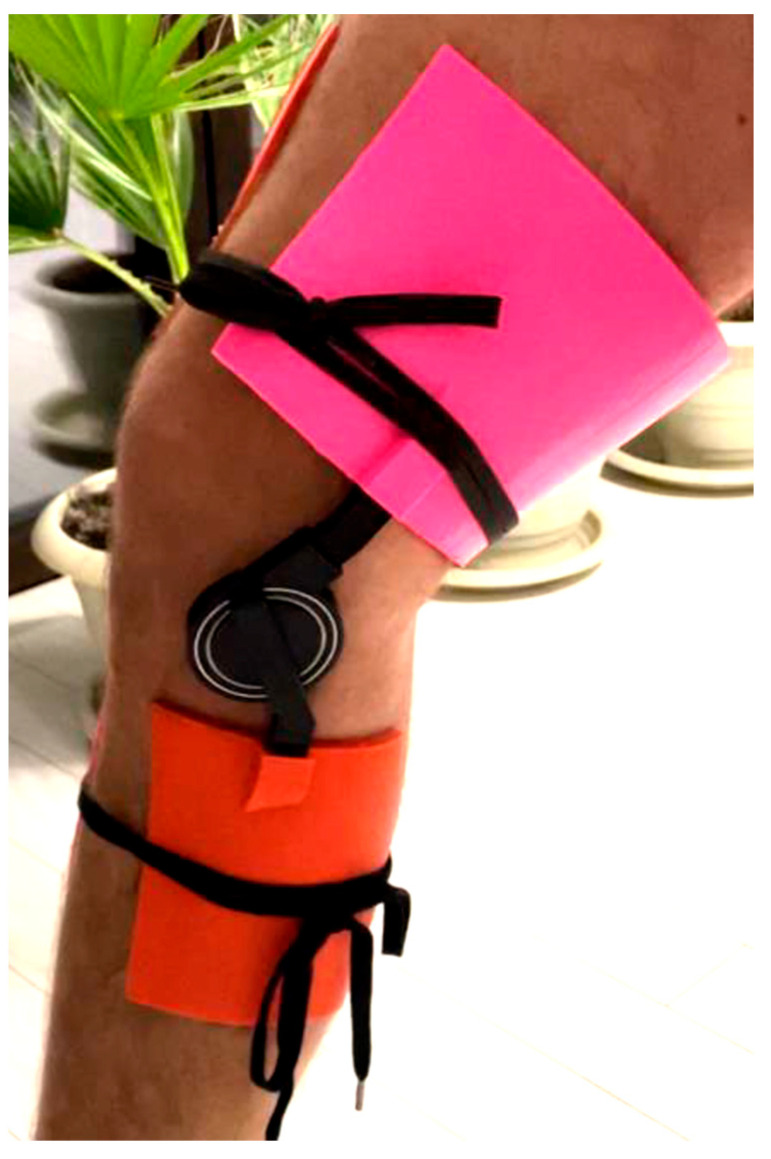
First iteration of the orthosis.

**Figure 5 polymers-16-01303-f005:**
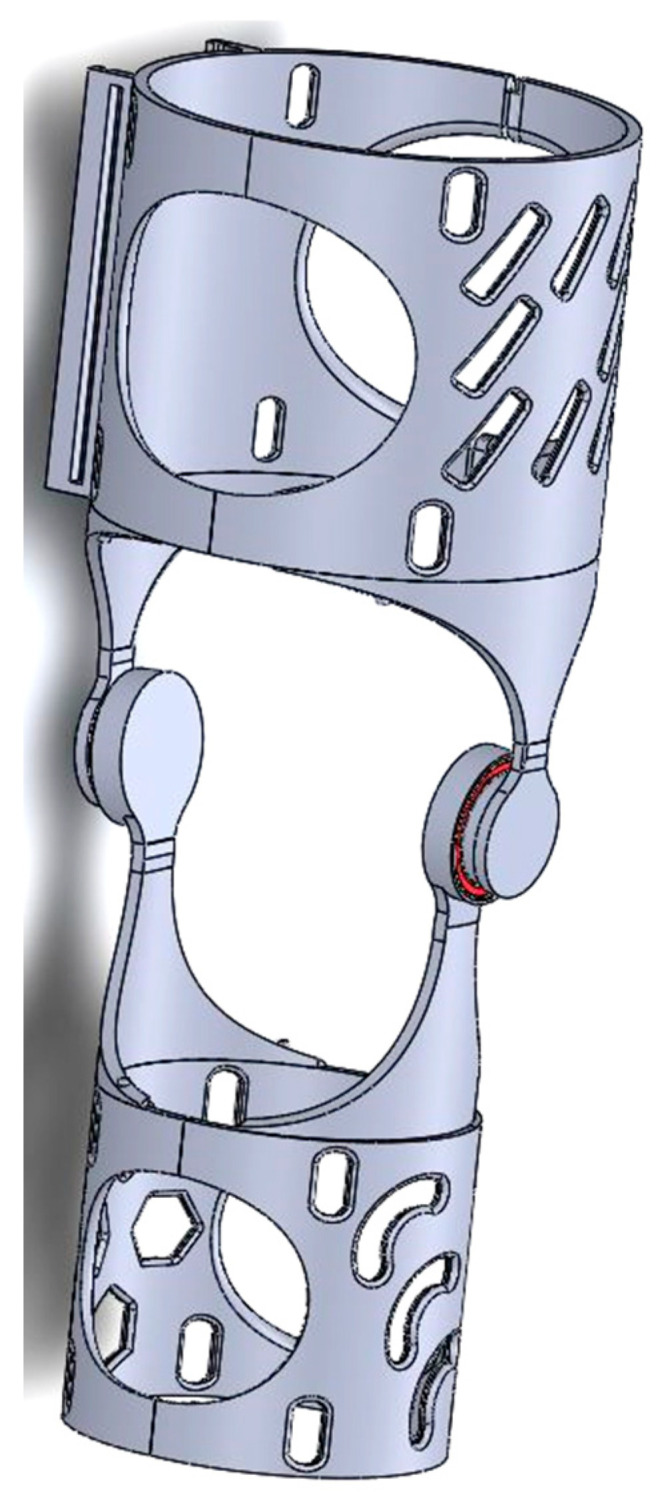
Final 3D model of the orthosis.

**Figure 6 polymers-16-01303-f006:**
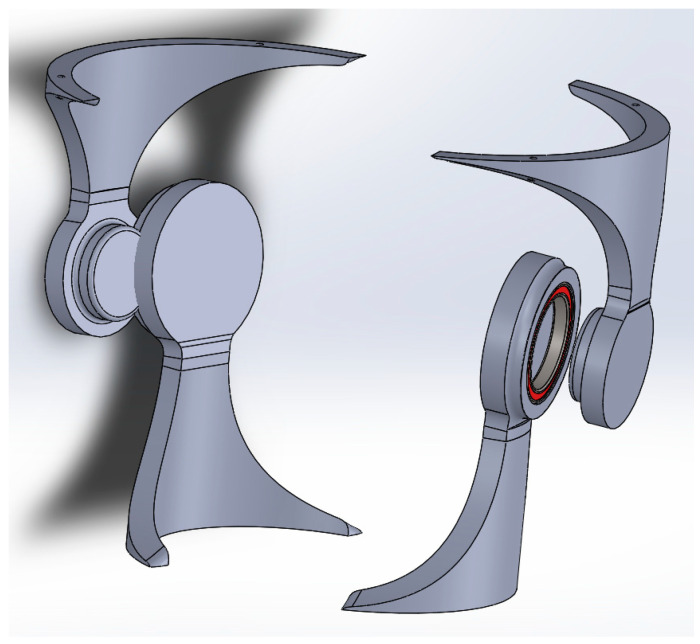
The components of the hinge of the orthosis.

**Figure 7 polymers-16-01303-f007:**
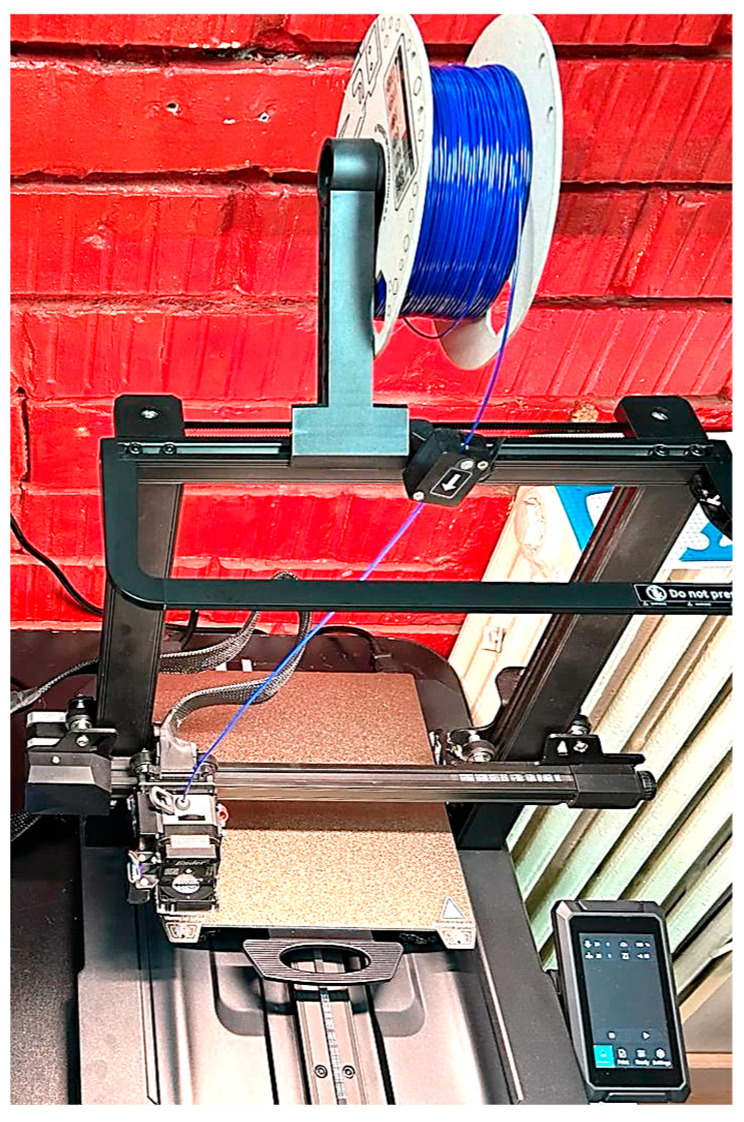
3D printing equipment is used to develop the physical model.

**Figure 8 polymers-16-01303-f008:**
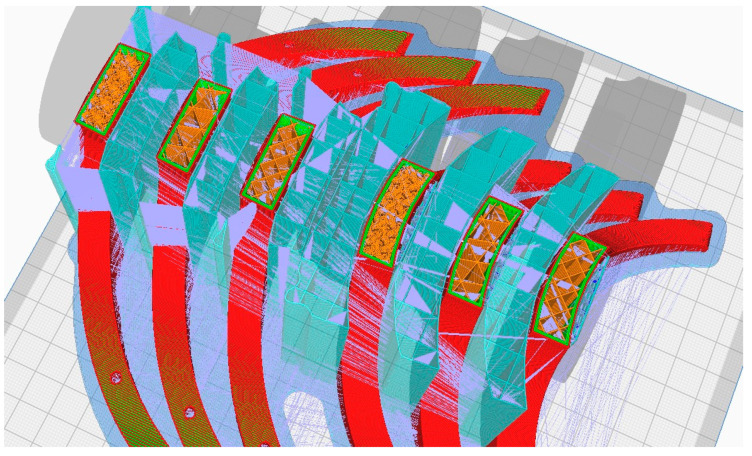
3D printing infill Pattern marked with orange color.

**Figure 9 polymers-16-01303-f009:**
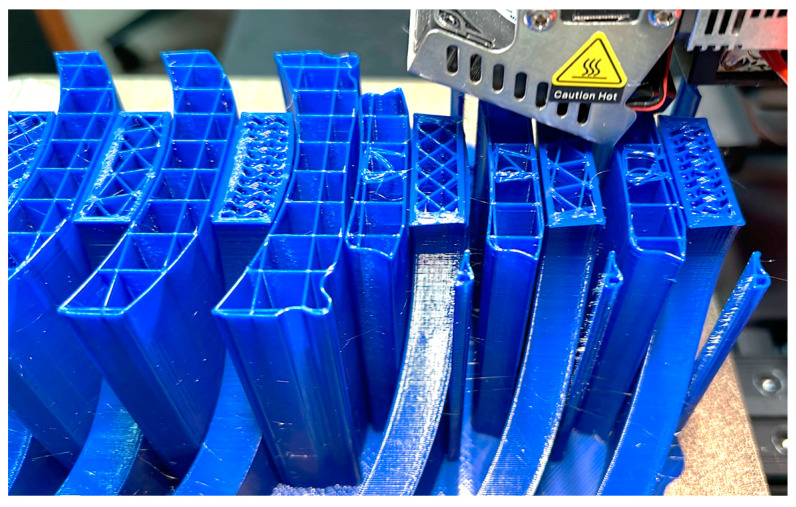
3D printing using rPET strategies, left to right (Grid/Triangles/Gyroid) without squares (support).

**Figure 10 polymers-16-01303-f010:**
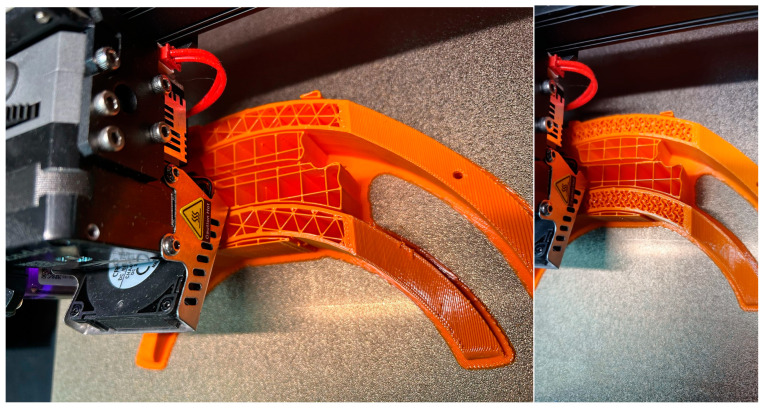
3D printing using PLA strategies, left to right (Grid/Triangles/Gyroid) without squares (support).

**Figure 11 polymers-16-01303-f011:**
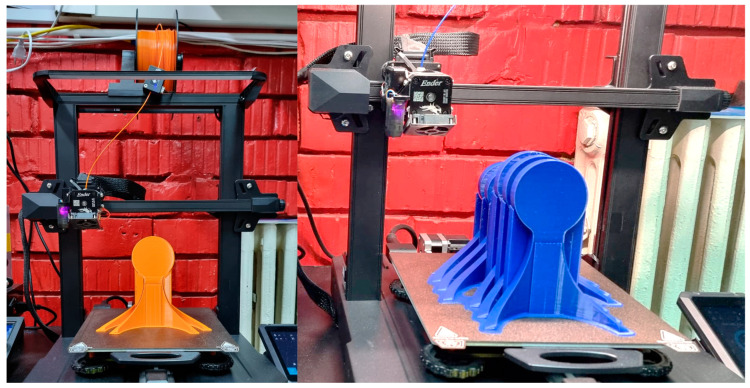
3D printing part using PLA (**left**) and rPET (**right**).

**Figure 12 polymers-16-01303-f012:**
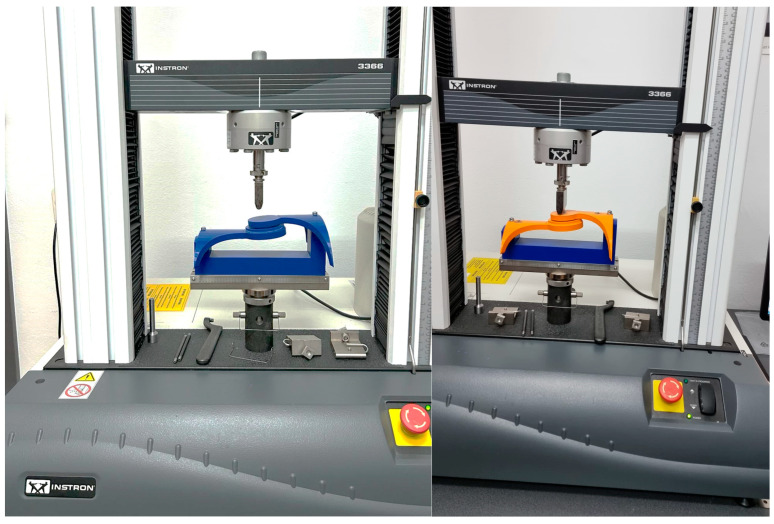
Instron^®^ 3366 Series (Norwood, MA, USA)—mechanical testing equipment; rPET (**left**) and PLA (**right**).

**Figure 13 polymers-16-01303-f013:**
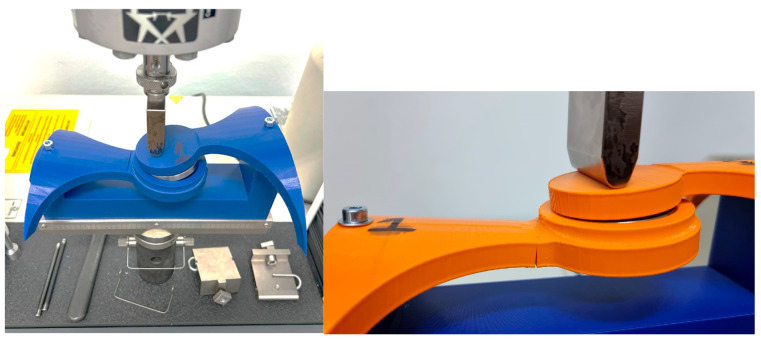
Testing **Specimen 1**—Infill Pattern -Grid—rPET (**left**) and PLA (**right**).

**Figure 14 polymers-16-01303-f014:**
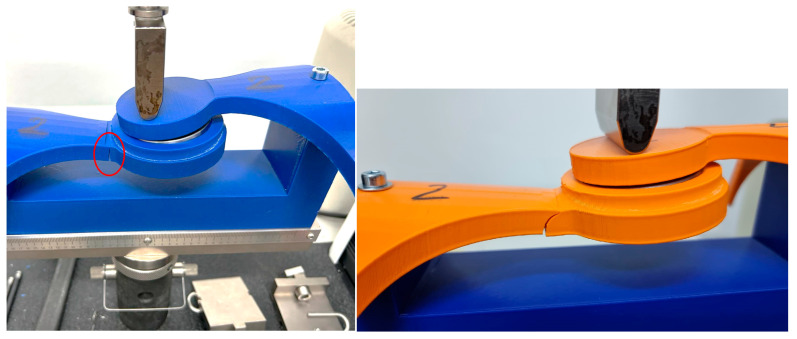
Testing **Specimen 2**—Infill Pattern—Gyroid—rPET (**left**) and PLA (**right**).

**Figure 15 polymers-16-01303-f015:**
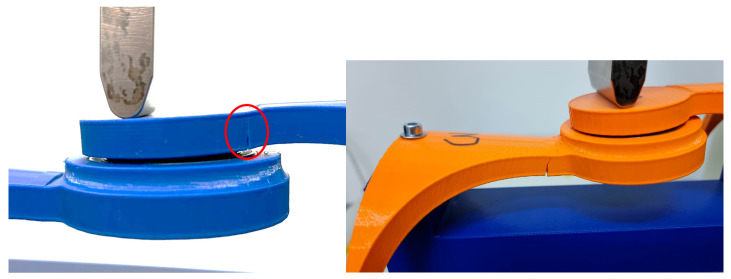
Testing **Specimen 3**—Infill Pattern—Triangles—rPET (**left**) and PLA (**right**).

**Figure 16 polymers-16-01303-f016:**
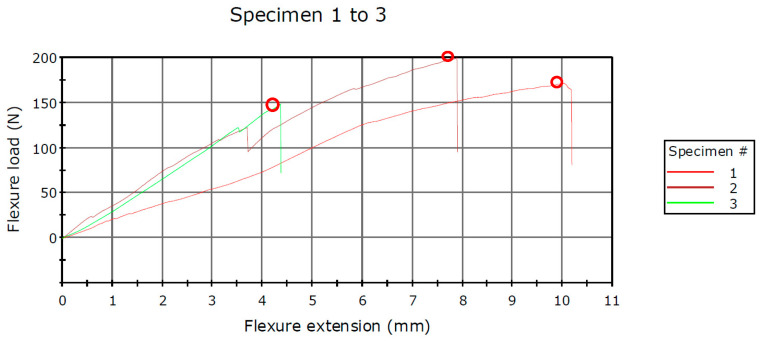
Schematic diagram of the force–displacement for the rPET samples.

**Figure 17 polymers-16-01303-f017:**
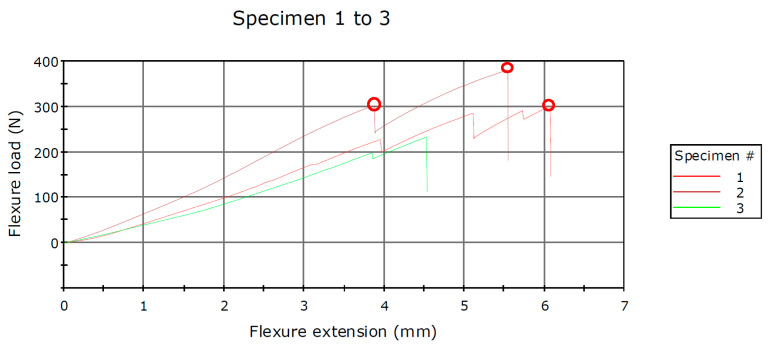
Schematic diagram of the force–displacement for the PLA samples.

**Figure 18 polymers-16-01303-f018:**
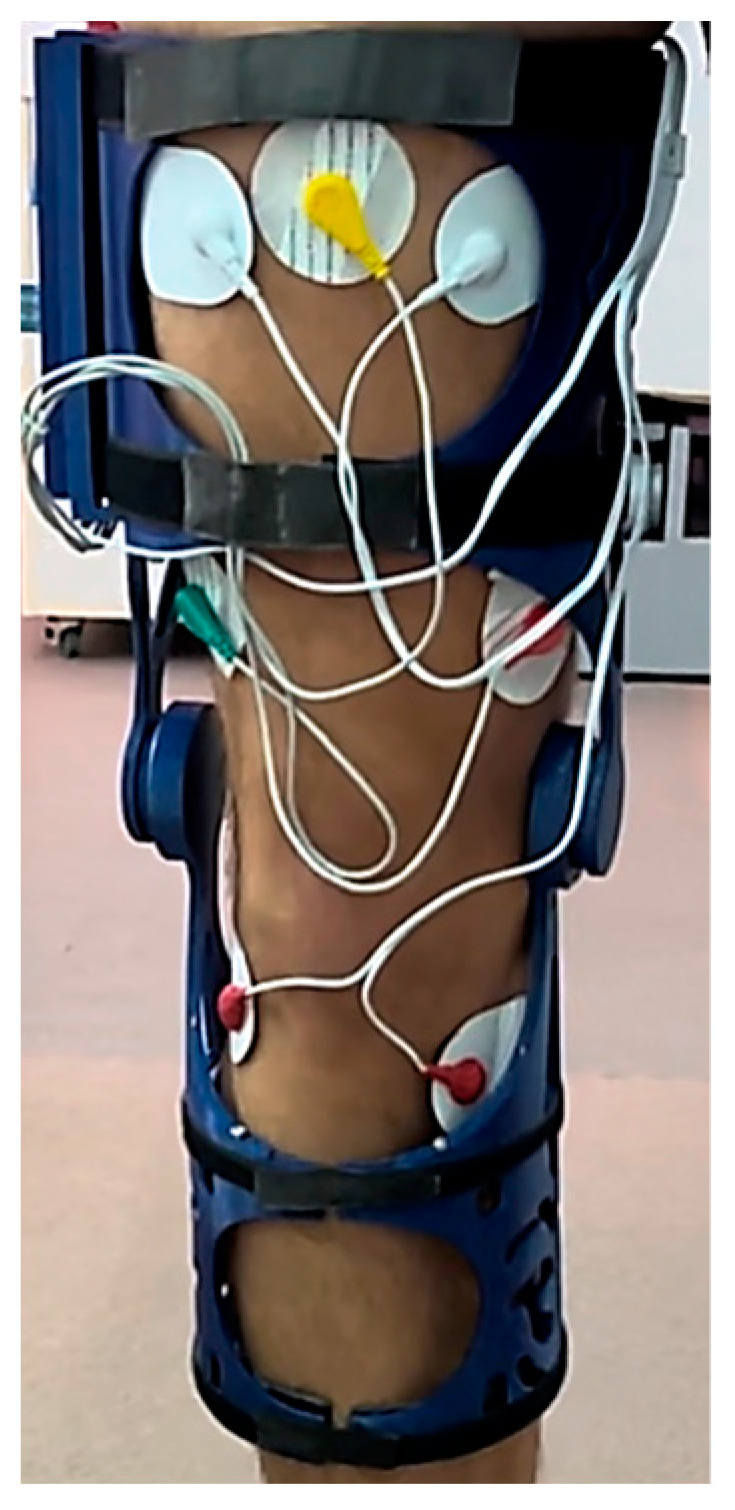
Physical model of the orthosis mounted on knee.

**Table 1 polymers-16-01303-t001:** The characteristics of the polymers were assessed utilizing information sourced from the FormFutura and filament datasheets [21].

Proprieties	ReFrom—rPET	EasyFil PLA
**Physical**		
Specific gravity	1.27 g/cc	1.24 g/cc
Melt flow rate	-	-
**Mechanical**		
Impact strength	7.2 KJ/m^2^	7.5 KJ/m^2^
Tensile strength	50 Mpa (MD)	110 Mpa (MD)
Tensile modulus	1940 Mpa (MD)	3310 Mpa (MD)
Elongation at break	120% (MD)	160% (MD)
Flexural strength	±70.6 Mpa	±55.2 Mpa
Flexural modulus	±2147.6 Mpa	±2392.5 Mpa

**Table 2 polymers-16-01303-t002:** The characteristics of the printing parameters used for 3D printing process.

	3D Printer: *Ender 3 Pro/Ender 3 S1 Pro*	Material: *rPET/PLA*	
PART	3D PRINTING STRATEGY	SETTINGS	VALUES
Lower Hinge	GRID/GYROID/TRIANGLES	INFILL DENSITY	30%
		LAYER HEIGHT	0.2
		PRINTING TEMPERATURE	230
		PRINTING SPEED	50
		BUILD PLATE ADHESION	Raft
	**3D printer** ***Ender 3 Pro/Ender 3 S1 Pro***	**Material: *rPET/PLA***	
**PART**	**3D PRINTING INFILL PATTERN**	**SETTINGS**	**VALUES**
Upper Hinge	GRID/GYROID/TRIANGLES	INFILL DENSITY	30%
		LAYER HEIGHT	0.2
		PRINTING TEMPERATURE	230
		PRINTING SPEED	50
		BUILD PLATE ADHESION	Raft

**Table 3 polymers-16-01303-t003:** Results of the bending analysis using rPET.

	Specimen Label	Flexure Load at Tensile Strength (N)	Flexure Extension at Tensile Strength (mm)
**1**	1	170.850	9.99170
**2**	2	198.954	7.88580
**3**	3	148.102	4.36130
**Coefficient of Variation**		14.75449	38.37660
**Maximum**		198.954	9.99170
**Mean**		172.636	7.41293
**Minimum**		148.105	4.36130
**Standard Deviation**		25.47162	2.84483

**Table 4 polymers-16-01303-t004:** Results of the bending analysis using PLA.

	Specimen Label	Flexure Load at Tensile Strength (N)	Flexure Extension at Tensile Strength (mm)
**1**	1	300.631	6.06696
**2**	2	379.680	5.53379
**3**	3	232.776	4.52523
**Coefficient of Variation**		24.15647	14.56627
**Maximum**		379.680	6.06696
**Mean**		304.362	5.37533
**Minimum**		232.776	4.52523
**Standard Deviation**		73.52320	0.78298

## Data Availability

Data are contained within the article.

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
