# Peer review of "Empowering Rehabilitation: Design and Structural Analysis of a Low-Cost 3D-Printed Smart Orthosis"

_polymers, 2024, doi:10.3390/polym16101303_

Round 1

Reviewer 1 Report

Comments and Suggestions for Authors

The article is interesting but requires a few minor corrections, especially regarding the figures.

Figure 1: A photo of the scanner with an indication of the sticker is not necessary for the scientific meaning of the work.

Figure 7. Shows 3D printing, which is a generally known process and does not add anything valuable to the work. Similarly, Figure 11.

Figure 12: Instead of showing photos from the tests, it seems better to show a loading diagram. This will explain the idea of the test and its reasonableness.

The load scale shown in Figures 16 and 17 should be the same. This will provide a better opportunity to compare the results. I also recommend marking the points on the force traces where orthosis has been destroyed.

The conclusions are somewhat broad and could be more specific about the future directions and potential for scale-up of the technology.

The article represents a significant contribution to the field of rehabilitation devices with its innovative approach to the design and fabrication of knee orthoses using 3D printing technology. However, enhancements in literature engagement, methodological detailing, and statistical robustness could elevate the work's impact and scientific rigor. With these improvements, the paper would be a strong candidate for publication in a journal focused on polymer research and its applications.

Author Response

We, the authors, would like to thank you for your time and also for evaluating our paper. 

Reviewer 2 Report

Comments and Suggestions for Authors

The manuscript titled “Empowering Rehabilitation: Design and Structural Analysis of a Low-Cost 3D Printed Smart Orthosis” was reviewed. The present study involves preparation and evaluation of a 3D (three-dimensional) printed knee orthosis. The basic idea of this study is promising; however, the manuscript requires major revisions.

Rationale for using PLA (polylactic acid) and PET (polyethylene terephthalate) to prepare 3D printed knee orthosis should be included in “Introduction” section.

First paragraph of “Introduction” section seems misplaced.  

Section “Introduction”, Page 2, “The production of orthoses using conventional methods is a time-consuming process.”, here, some conventional methods should be mentioned.

Section “2. Prototype design and manufacture”, Page 8, “To optimize printing outcomes with rPET and PLA material, it might be imperative to enact certain adjustments to the printing parameters, see Table 2, encompassing parameters such as layer height, print speed, and temperature settings, to ensure optimal print quality and adherence to design specifications”, any screening results of these printing parameters to attain an optimized knee orthosis are not mentioned in the manuscript.

Detailed methodology of mechanical test should be included.

Figure 18 and Section “2. Prototype design and manufacture”, it has been mentioned that some sensors were integrated in the knee orthosis, however, details about optimum functionality/working of these sensors are missing. 

Section “3. Results”, Page 13, “The development of a tailored mobile application utilizing MIT App Inventor, coupled with the establishment of robust Bluetooth communication between the application and the knee brace system, posed a substantial challenge. The application served as a user- centric interface, facilitating the real-time display of sensor data, management of system settings, and seamless user interaction.” Any details of mobile application and real-time display of sensor data are not mentioned anywhere in the manuscript. These details should be included.

Majority of the results are included in section “2. Prototype design and manufacture” and “Results” section seems more like “Discussion”. The data arrangement/organization should be improved.

Overall, the results seem incomplete/missing; the presentation of results/data should be improved.

Figure 16 and 17, details of specimens 1,2,3 should be mentioned.

Standard deviation is missing at some points in the manuscript; should be included.

Full forms of all abbreviations should be included.

Comments on the Quality of English Language

Linguistic and typographical errors should be removed.

Author Response

First, we the authors would like to thank you for your time and also for evaluating our paper. 

Round 2

Reviewer 1 Report

Comments and Suggestions for Authors

Even though not all the suggested corrections have been implemented, the article is suitable in its current form.

Reviewer 2 Report

Comments and Suggestions for Authors

The revised manuscript may be accepted for publication 

Comments on the Quality of English Language

Minor typographic corrections required which can be addressed in proofs